# Interplay between structural hierarchy and exciton diffusion in artificial light harvesting

Björn Kriete[1], Julian Lüttig[2], Tenzin Kunsel [1], Pavel Malý[2], Thomas L.C. Jansen [1], Jasper Knoester[1], Tobias Brixner [2,3] & Maxim S. Pshenichnikov[1]*

Unraveling the nature of energy transport in multi-chromophoric photosynthetic complexes is essential to extract valuable design blueprints for light-harvesting applications. Long-range exciton transport in such systems is facilitated by a combination of delocalized excitation wavefunctions (excitons) and exciton diffusion. The unambiguous identification of the exciton transport is intrinsically challenging due to the system's sheer complexity. Here we address this challenge by employing a spectroscopic lab-on-a-chip approach: ultrafast coherent two-dimensional spectroscopy and microfluidics working in tandem with theoretical modeling. We show that at low excitation fluences, the outer layer acts as an exciton antenna supplying excitons to the inner tube, while under high excitation fluences the former converts its functionality into an exciton annihilator which depletes the exciton population prior to any exciton transfer. Our findings shed light on the excitonic trajectories across different sub-units of a multi-layered artificial light-harvesting complex and underpin their great potential for directional excitation energy transport.

[1] University of Groningen, Zernike Institute for Advanced Materials, Nijenborgh 4, 9747 AG Groningen, The Netherlands. [2] Institut für Physikalische und Theoretische Chemie, Universität Würzburg, Am Hubland, 97074 Würzburg, Germany. [3] Center for Nanosystems Chemistry (CNC), Universität Würzburg, Theodor-Boveri-Weg, 97074 Würzburg, Germany. *email: M.S.Pchenitchnikov@rug.nl

Many natural photosynthetic complexes utilize light-harvesting antenna systems that enable them to perform photosynthesis under extreme low light conditions only possible due to remarkably efficient energy transfer[1]. The success of natural systems, such as the multi-walled tubular chlorosomes of green sulfur bacteria, relies on the tight packing of thousands of strongly coupled molecules[2]. This arrangement facilitates the formation of collective, highly delocalized excited states (Frenkel excitons) upon light absorption as well as remarkably high exciton diffusivities[3]. Understanding the origin of the delocalized states and tracking energy transport throughout the entire complex hierarchical structures of multi-chromophoric systems—from the individual molecules, over individual sub-units all the way up to the complete multi-layered assembly—is vital to unravel nature's highly successful design principles.

In reality, however, natural systems are notoriously challenging to work with as they suffer from sample degradation once extracted from their stabilizing environment and feature inherently heterogeneous structures[4,5], which disguises relations between supramolecular morphology and excitonic properties. In this context, a class of multi-layered, supramolecular nanotubes holds promise as artificial light-harvesting systems owing to their intriguing optical properties and structural homogeneity paired with self-assembly capabilities and robustness[6–8]. Previous studies have demonstrated the potential of these systems as quasi-one-dimensional long-range energy transport wires[9–13], where the dependence of the transport properties on the hierarchical order as well as dimensionality of the respective system is a re-occurring topic of great interest[14–16]. Nevertheless, even in these simpler structures the delicate interplay between individual sub-units of the supramolecular assembly hampers the unambiguous retrieval of exciton transport dynamics.

Recent studies have focused on reducing the complexity of multi-layered, supramolecular nanotubes and thereby essentially uncoupling individual hierarchical units, i.e., the inner and outer layer of the assembly by oxidation chemistry[7,8,17,18]. In addition, Eisele et al. have demonstrated flash-dilution as an elegant tool to selectively dissolve the outer layer to obtain an unobscured view on the isolated inner layer[7,14]. Nevertheless, the rapid recovery of the initial nanotube structure within a few seconds impedes studies more elaborate than simple absorption—for instance, time-resolved spectroscopy—to probe exciton dynamics. A strategy that is capable to alleviate these limitations relies on microfluidics[19], which in recent years has successfully been implemented to manipulate chemical reactions in real time[20] or to steer self-assembly dynamics[21,22]. In particular, combinations of microfluidics and spectroscopy including steady-state absorption[23], time-resolved spectroscopy[24,25], and coherent two-dimensional (2D) infrared spectroscopy[26] have received considerable attention. In this framework, microfluidics bridges the gap between controlled modifications of the sample on timescales of microseconds to minutes with ultrafast processes on timescales down to femtoseconds.

In parallel with these developments, electronic 2D spectroscopy[27] has evolved to a state-of-the-art tool for investigation of exciton dynamics in multi-chromophoric and other complex systems with significant inputs from both theory[28–33] and experiment[34–42]. Recently, a fifth-order 2D spectroscopic technique has been demonstrated to be capable of resolving exciton transport properties by directly probing mutual exciton–exciton interactions (hereafter denoted as EEI)[43].

In this paper, we identify the dynamics of excitons residing on different sub-units of a multi-walled artificial light-harvesting complex. Disentangling the otherwise complex response is made possible by successfully interfacing EEI2D spectroscopy with a microfluidic platform, which provides spectroscopic access to the simplified single-walled nanotubes. We show that experimental EEI2D spectra, together with extensive theoretical modeling, provide an unobscured view on exciton trajectories throughout the complex supramolecular assembly and allows one to obtain a unified picture of the exciton dynamics (Table 1).

## Results and discussion

**Microfluidic flash-dilution.** We investigate double-walled C8S3-based nanotubes (chemical structure shown in Fig. 1a) whose linear absorption spectrum (Fig. 1b, black solid line) comprises two distinct peaks that have been previously assigned to the outer (589 nm, $\omega_{outer}$ ~17,000 cm$^{-1}$) and inner layer (599 nm, $\omega_{inner}$ ~16,700 cm$^{-1}$) of the assembly[7,17]. The spectral red-shift of ~80 nm (~2400 cm$^{-1}$) and a tenfold spectral narrowing relative

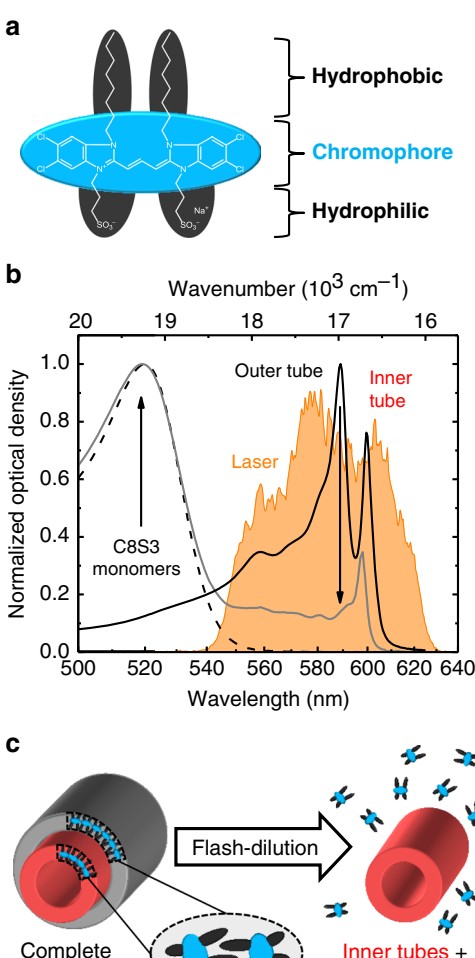

**Fig. 1** Investigated system and absorption spectra before and after flash-dilution. **a** Molecular structure of the C8S3 molecule with the chromophore and functional side-groups highlighted in light blue and dark gray, respectively. **b** Linear absorption spectra of neat nanotubes (black solid line), isolated inner tubes (gray solid line), and dissolved monomers (black dashed line) in methanol. The laser excitation spectrum (orange) is shown for comparison. Arrows indicate spectroscopic changes upon flash-dilution. **c** Schematic representation of the flash-dilution process that selectively strips the outer tube, while leaving a sufficient share of the inner tubes intact. The decreased amplitude of the peak at ~600 nm indicates partial dissolution of inner tubes. The dissolved monomers contribute to a broad absorption band around ~520 nm, which is not covered by the excitation spectrum and, thus, has no consequences for ultrafast spectroscopy (Supplementary Note 2 and Supplementary Fig. 2)

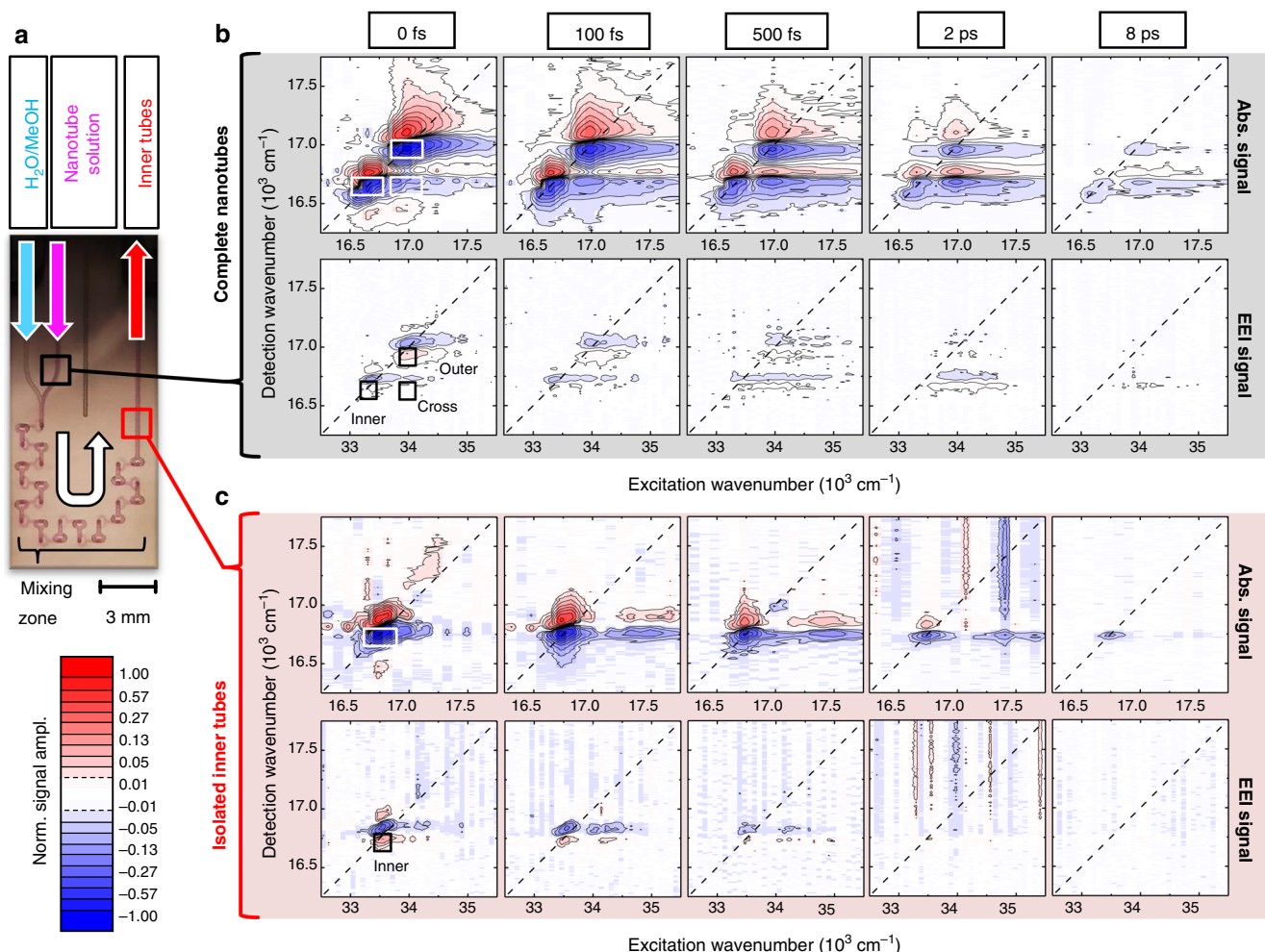

**Fig. 2** Absorptive and EEI 2D spectra recorded before and after microfluidic flash-dilution. **a** A photograph of the cuvette for microfluidic flash-dilution via mixing of neat nanotube solution and a diluting agent (50:50 mixture by volume of $H_2O$ and methanol). Arrows indicate the flow direction of the solvents. (**b**) and (**c**) Representative absorptive 2D and EEI2D spectra at selected waiting times measured for complete (panel b; shaded gray) and isolated inner tubes (panel c; shaded red). The spectra were normalized to the maximum absolute amplitude at 0 fs waiting time. The signal amplitude is depicted on a color scale ranging from −1 to 1, with increments at 0.83, 0.57, 0.4, 0.27. 0.19, 0.13, 0.08, 0.05, 0.03, and 0.01 to ensure visibility of all peaks at all waiting times. Contour lines are drawn as specified in the color bar except for the lower signal levels for isolated inner tubes. Negative and positive features in the absorptive 2D spectra refer to ground-state bleach/stimulated emission (GSB/SE) and excited-state absorption (ESA) signals, respectively. In the EEI2D spectra the signal signs are opposite, which is caused by the two additionally required interactions with the incident light fields and the associated factor of $i^2 = −1$ within the perturbation expansion[28,43]. The direct comparability of the absorptive and EEI signals is ensured, because both signals are recorded under identical conditions, as they are emitted in the same phase-matched direction and captured simultaneously. Diagonal lines (dashed) are drawn at $\omega_{excitation} = \omega_{detection}$ and $\omega_{excitation} = 2\omega_{detection}$ for absorptive 2D and EEI2D spectra, respectively. White and black rectangles depict the regions of interest in which the signal was integrated to obtain the transients (Supplementary Note 3 and Supplementary Table 1). The exciton density corresponds to one exciton per ~20 and ~60 individual molecules for isolated inner tubes and complete nanotubes, respectively. Additional 2D spectra for low exciton densities are presented in Supplementary Fig. 3 and Supplementary Fig. 4

to the monomer absorption is typical for J-aggregation[6]. The magnitude of these effects evidences strong intermolecular couplings, which are essential for the formation of delocalized excited states. A number of weaker transitions at the blue flank of the nanotube spectrum were previously ascribed to the complex molecular packing into helical strands[44] with two molecules per unit cell[7]. It has previously been shown that the two main transitions as well as one of the weaker transitions at ~571 nm (~17,500 cm$^{-1}$) are polarized parallel, while all remaining transitions are polarized orthogonal to the nanotube's long axis[17]. The nanotubes preferentially align along the flow in the sample cuvette due to their large aspect ratio (outer diameter ~13 nm, length several micrometers). As a result, the laser pulses polarized along the flow selectively excite transitions that are polarized parallel to

the long axis of the nanotube, i.e., predominantly the two main transitions.

Controlled destruction of the outer layer (Fig. 1c) was achieved in a microfluidic flow-cell (Fig. 2a) by mixing nanotube solution with a diluting agent (50:50 mixture by volume of $H_2O$ and methanol). Continuous dissolution is evident from the absence of the outer tube absorption peak, while the peak associated with the inner tube is retained (Fig. 1b, gray line), which corroborates the 1-to-1 assignment of these peaks to the inner and outer tube. Simultaneously, a new absorption peak around 520 nm (~19,200 cm$^{-1}$) indicates an increase in monomer concentration that formerly constituted the outer layer. We use this peak to estimate the concentration of molecules that remains embedded in the inner tubes upon flash-dilution (Supplementary Note 1 and Supplementary Fig. 1).

**Exciton–exciton interaction 2D (EEI2D) spectroscopy.** A set of representative 2D spectra obtained for complete nanotubes and isolated inner tubes at different waiting times $T$ and the excitation axis expanded to more than twice the fundamental frequency $2\omega$ are shown in Fig. 2b, c. We will refer to the $\omega$ and $2\omega$ regions as absorptive 2D and EEI2D spectra, respectively. It has previously been shown that the $2\omega$ region is dominated by signals that encode exciton–exciton interactions, e.g., exciton–exciton annihilation (EEA)[43,45]. Hence, the structure and dynamics of the EEI2D spectra allow tracing the annihilation of two excitons with their trajectories encoded in the amplitude and spectral position of the respective peak as functions of the waiting time $T$.

For complete nanotubes, the absorptive 2D spectra at early waiting times are characterized by two pairs of negative ground-state bleach/stimulated emission (GSB/SE) and positive excited-state absorption (ESA) diagonal peaks with the low- and high-energy pair associated with the inner tube and outer tube, respectively (Fig. 2b). For later waiting times, a cross peak clearly emerges below the diagonal, for which again GSB/SE and ESA features can be identified; these data are in line with previous publications[14,46]. A cross peak above the diagonal can also be identified; however, it has a low amplitude because of thermally activated ($\Delta E \approx 300\,\mathrm{cm}^{-1}$) energy transfer from the inner to the outer tube and its partial spectral overlap with ESA of the inner tube. The EEI2D spectra essentially mirror the absorptive 2D spectra evidencing intensive exciton–exciton interactions on each individual tube (diagonal peaks) as well as between the tubes (cross peaks).

Upon microfluidic flash-dilution of the outer wall, the 2D spectra simplify to a single pair of GSB/SE and ESA peaks

originating from the isolated inner tubes at an excitation frequency of ~16,700 cm⁻¹ (Fig. 2c). Expectedly, neither a diagonal peak showing the presence of the outer tube nor a cross peak indicating inter-layer exciton transfer is detected. The absence of the outer tube spectrally isolates weak cross peaks at a detection frequency of ~16,700 cm⁻¹ and excitation frequencies of ~17,500 cm⁻¹ and ~35,000 cm⁻¹ in the absorptive 2D and EEI2D spectra, respectively. These peaks are linked to the blue-shifted transition in the nanotube absorption (Fig. 1b) and are not relevant for the further analysis due to their small amplitude (Supplementary Note 4 and Supplementary Fig. 5).

In the further analysis, we will focus on the GSB/SE components of the absorptive and EEI signals corresponding to the diagonal outer tube, diagonal inner tube and their low-frequency cross peak, from which we extract the amplitudes as a function of the waiting time for all measured exciton densities by integrating the signal in the rectangles (250 cm⁻¹ along the excitation and 100 cm⁻¹ along the detection axis; depicted in Fig. 2b, c; Supplementary Table 1). The GSB/SE signals contain information on the creation of excitons residing on different, spatially separated domains followed by EEA due to exciton diffusion.

**Exciton dynamics of isolated inner tubes.** We begin our analysis with the isolated inner nanotubes (Fig. 3a). Increasing the exciton density leads to a progressively growing amplitude of the absorptive signal at early waiting times with the onset of saturation at the highest exciton density of 1 exciton per ~20

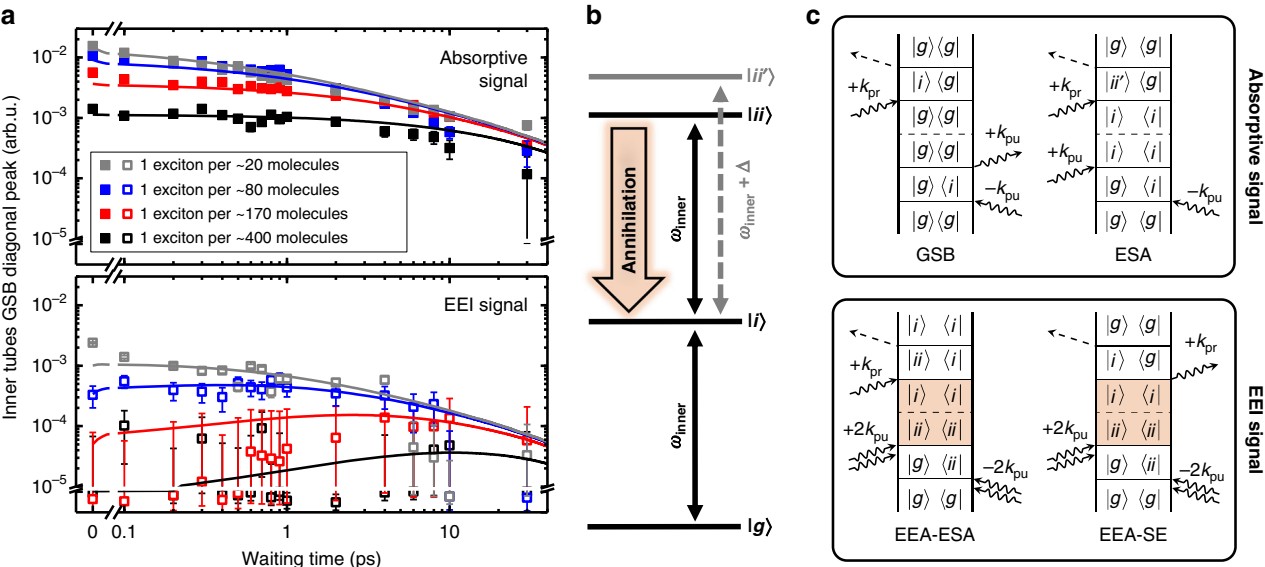

**Fig. 3** Absorptive and EEI transients of isolated inner tubes. **a** Log–log plot of the absorptive (upper panel, solid squares) and EEI (lower panel, open squares) GSB/SE transients for isolated inner tubes for different exciton densities. The transients were obtained by integrating the signal in the rectangular regions of interest shown in Fig. 2c; the panels are drawn with the same scaling to emphasize their direct comparability, which is one of the constraints in the Monte Carlo simulations (*vide infra*). The sign of the EEI responses was inverted for the ease of comparison. The error bars refer to the detection noise level in the experiment (Supplementary Note 3). The solid lines depict the results from Monte Carlo simulations of the exciton dynamics on isolated inner tubes. The amplitude (vertical) scaling between experimental and simulated data is preserved, i.e., for each signal (absorptive and EEI) a single scaling factor was used for *all* simulated transients. **b** Energy level diagram of the isolated inner nanotubes with the electronic ground state ($|g\rangle$) and the one- ($|i\rangle$) and bi-exciton ($|ii\rangle$) states ($i$ stands for the *i*nner tube). Optical transitions are marked by vertical black arrows with the corresponding frequency $\omega_{inner}$. The blue-shifted one- to two-exciton transition within the same excited domain ($|ii'\rangle$, dashed gray arrow; refs. [47,48]) is shown for comparison. Bold arrow: annihilation channel from the bi-excitonic state. **c** Representative set of rephasing double-sided Feynman diagrams, which contribute to the absorptive ($\omega_{inner} \rightarrow \omega_{inner}$; upper panel) and EEI ($2\omega_{inner} \rightarrow \omega_{inner}$; bottom panel) diagonal peaks of isolated inner tubes. In the diagrams time flows from the bottom to the top during which the interactions with the laser pulses are indicated by arrows. The dashed line indicates propagation during the waiting time $T$. The double interaction with each of the two pump pulses can create a population of the ground state, a one-exciton state or a bi-exciton state, which are subsequently probed by GSB ($|g\rangle \rightarrow |i\rangle$), SE ($|i\rangle \rightarrow |g\rangle$ or $|ii\rangle \rightarrow |i\rangle$) or ESA ($|i\rangle \rightarrow |ii\rangle$ or $|i\rangle \rightarrow |ii'\rangle$). The process of exciton–exciton annihilation (EEA) is shaded in orange

molecules (Fig. 3a, upper panel). Furthermore, the transients decay faster at longer waiting times which is a typical fingerprint for EEA encoded in the EEI signal.

In order to dissect the contributions to the EEI signal, we describe the isolated inner tubes as a three-level system (Fig. 3b, c). The detection frequency selection allows one to distinguish between the bi-exciton state of two separate singly excited domains ($\omega_{inner}$) and the one- to two-exciton transition within the same excited domain ($\omega_{inner} + \Delta$)[47,48]. For J-aggregates, the latter occurs blue-shifted relative to the ground-state to one-exciton transition ($\Delta > 0$) as a consequence of Pauli repulsion between excitons[49], as two excitations cannot reside on the same molecule. This effective repulsion between Frenkel excitons dominates Coulomb interactions between them if the difference in the permanent dipole between the ground and excited states considered is zero. EEA opens a relaxation channel between the $|ii\rangle$ and $|i\rangle$ states[28,31,32,43]. Next to the re-appearance of the otherwise mutually annulled Feynman diagrams, this leads to new diagrams as shown in Fig. 3c, which in turn results in the emergence of the EEI signal (Fig. 3a). The complete set of the relevant Feynman diagrams for the inner diagonal peak is provided in Supplementary Note 5.1 and Supplementary Figs. 6 and 7.

At low exciton densities the EEI signal is barely detectable at the noise background (Fig. 3a, black squares), while higher exciton densities lead to the rapid emergence of the EEI signal. For sparse exciton populations a delayed formation of the maximum annihilation signal is glimpsed at a waiting time of ~8 ps (Fig. 3a, red squares), because excitons must diffuse toward each other prior to annihilation. This maximum is gradually shifting toward earlier waiting times for higher exciton densities, as a shorter and shorter period is required before individual excitons meet and annihilate. For the highest exciton density, the maximum EEI signal occurs at essentially zero waiting time, as excitons annihilate with virtually no time to diffuse. These features qualitatively agree with predictions of analytical models for diffusion-assisted bi-excitonic annihilation in one and two dimensions[43,50–52]. However, the quantitative description is prevented by the fact that the isolated inner tubes fall in neither category, as the underlying molecular structure shows characteristics of both: helical molecular strands (1D) mapped onto the surface of a cylinder (2D).

We analyze the experimental data using Monte Carlo (MC) simulations, where we describe the exciton dynamics in a combined framework of diffusive exciton hopping and exciton–exciton interactions[43,53–55]; see Methods section, Supplementary Note 6.1 and Supplementary Table 2. For comparison with experiment, we obtain the amplitude of the absorptive signal by counting the total number of excitons at time $T$ in the MC simulations, whereas for the EEI signal only excitons that have participated in at least one annihilation event are calculated (Supplementary Note 6.2 and Supplementary Table 3). The latter occurs if two excitons approach each other closer than the annihilation radius, which we define as the cut-off distance for exciton–exciton interactions (Supplementary Note 6.3 and Supplementary Fig. 11). We find excellent agreement of the experimental data (Fig. 3a, squares) and the simulated curves (Fig. 3a, solid lines) by global adjustment of only two parameters: the exciton diffusion of $D_{2D} \sim 5.5 \, nm^2 \, ps^{-1}$ (equivalent to 10 molecules $ps^{-1}$ given the molecular grid in the MC simulations) and the exciton annihilation radius of 3 molecules; an overview of all parameters is given in the Methods section. The 2D diffusion constant was obtained via the mean square exciton displacement ($<x^2> = 4D_{2D}\tau$; Supplementary Note 6.4 and Supplementary Fig. 12) in the annihilation-free case. Our simulations also revealed that pure two-excitonic annihilation, where each exciton can only participate in a single annihilation event, is not appropriate to describe the data set in its entirety. Instead, we find that already the lowest experimental exciton density requires a multi-exciton description, where according to our MC simulations ~ 30% of the excitons are involved in at least two annihilation events (Supplementary Note 6.5 and Supplementary Fig. 13). Evidence for these processes is encoded in even higher order (i.e., at least seventh-order) 2D spectra, which have indeed been observed experimentally (Supplementary Note 5.3, Supplementary Note 7 and Supplementary Figs. 10 and 16).

**Cross-peak dynamics of complete nanotubes**. Now we are in position to elucidate the changes of the exciton dynamics induced by the presence of the outer layer, which involve both intra- and inter-tube exciton interactions. In analogy with the isolated inner tubes, the diagonal peaks in the EEI2D spectra for the inner and outer tube reveal annihilation of excitons that were initially planted on the same layer (Fig. 4). The salient differences of the dynamics of the complete nanotubes compared to the isolated inner tubes arise from the inter-tube exciton transfer (ET), which is evident from the mere existence of the cross peaks in the absorptive and EEI2D spectra (Fig. 2b). These peaks reveal coupling of the individual layers, which leads to an inter-layer exchange of excitons on a sub-ps timescale. Hence, the additional information on specific exciton trajectories including inter-layer ET and EEA is encoded in the absorptive and EEI cross peaks, whose maxima are found to gradually shift to earlier waiting times for increasing exciton densities (Fig. 5a), while their amplitudes saturate for the highest exciton density similarly to the trend found for the inner tubes.

Dissecting the individual contributions to the EEI cross peak is crucial to unravel the effect of the multi-layered structure for the observed exciton dynamics, yet intrinsically challenging due to the wealth of possible exciton trajectories. Therefore, we limit our analysis to the EEI cross peak linking the creation of two excitons on the outer layer with the detection of a single exciton on the inner layer, i.e., $2\omega_{outer} \rightarrow \omega_{inner}$ (see Supplementary Note 5.2 and Supplementary Figs. 8 and 9 for the corresponding Feynman diagrams). We consider this process dominant for two reasons: first, the total (initial) number of excitons on the outer tube is significantly larger as its absorption cross-section is a factor of ~2 higher than for the inner tubes and, second, at early waiting times the majority of ET events occurs from the outer to the inner tube (i.e., downhill in energy). We extend the three-level system of the isolated inner tubes by also including the one- and bi-excitonic states of the outer tube as $|o\rangle$ and $|oo\rangle$ (Fig. 5b). We assume that EEA can only occur from bi-excitonic states populating the same tube ($|oo\rangle$ and $|ii\rangle$) and not from the mixed population state $|oi\rangle$, which describes two single excitons residing on spatially separated domains on each tube. This assumption is based on the fact that due to the wall separation of ~3.5 nm the inter-tube dipole–dipole interactions that are responsible for EEA are negligibly small compared to the dipole–dipole interactions within the same tube[7,8]. Nevertheless, we consider the mixed state as one of the pathways via which excitons from the outer tube bi-excitonic state can be transferred to the inner tube bi-excitonic state prior to any EEA.

At zero waiting time, neither an absorptive nor an EEI cross peak is expected, since excitons have no time to undergo ET and EEA. For finite waiting times, however, the EEI cross peak is dominated by processes that simultaneously include EEA and ET. EEA can occur via two annihilation channels: (1) ET of two excitons created on the outer tube followed by EEA on the inner tube (Fig. 5b; highlighted in blue), or (2) EEA on the outer tube followed by ET of the surviving exciton to the inner tube (Fig. 5b;

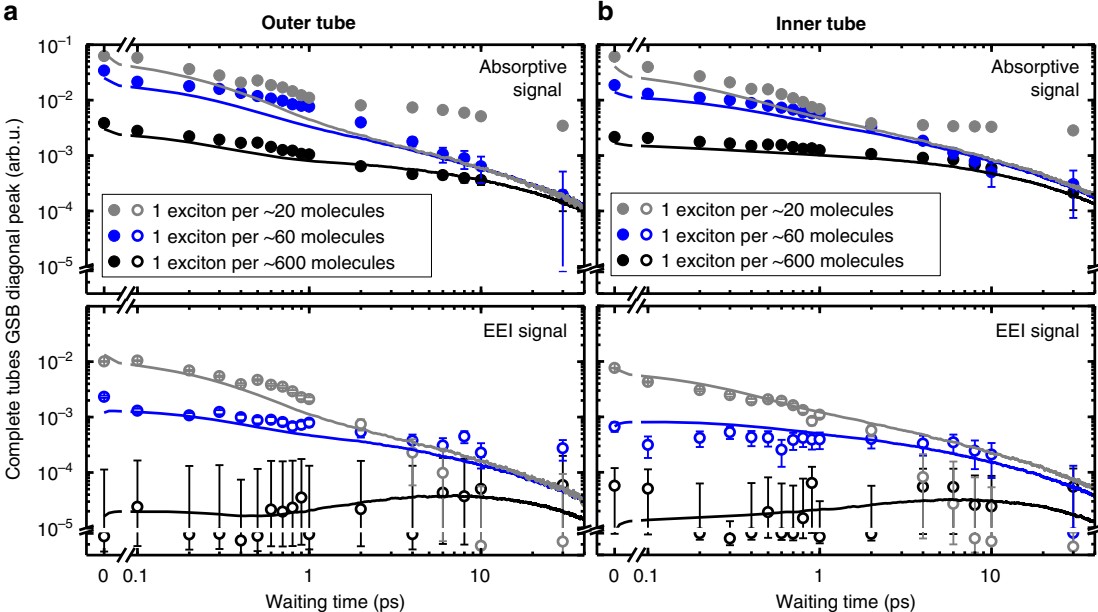

**Fig. 4** Absorptive and EEI transients of both layers of complete nanotubes. Log–log plots of the absorptive (upper panels, solid circles) and EEI (lower panels, open circles) GSB/SE transients for (a) outer and (b) inner tube diagonal peaks at different exciton densities. The transients were obtained by integrating the signal in the rectangular regions of interest shown in Fig. 2b. The panels are drawn with the same scaling to emphasize their direct comparability, as both are derived from the same signal. The error bars refer to the detection noise level in the experiment (Supplementary Note 3). The solid lines depict the results from Monte Carlo simulations of the exciton dynamics on isolated inner tubes. The amplitude (vertical) scaling between experimental and simulated data is preserved, i.e., for each signal (absorptive and EEI) a single scaling factor was used for *all* simulated transients. The sign of the EEI responses was inverted for the ease of comparison. Deceleration of the transient dynamics at $T > 2$ ps for the highest exciton density (1 exciton per ~20 molecules) is caused by transient heating of the nanotubes and a few surrounding water layers as a result of the energy released by exciton annihilation events (Supplementary Note 8 and Supplementary Figs. 17 and 18)

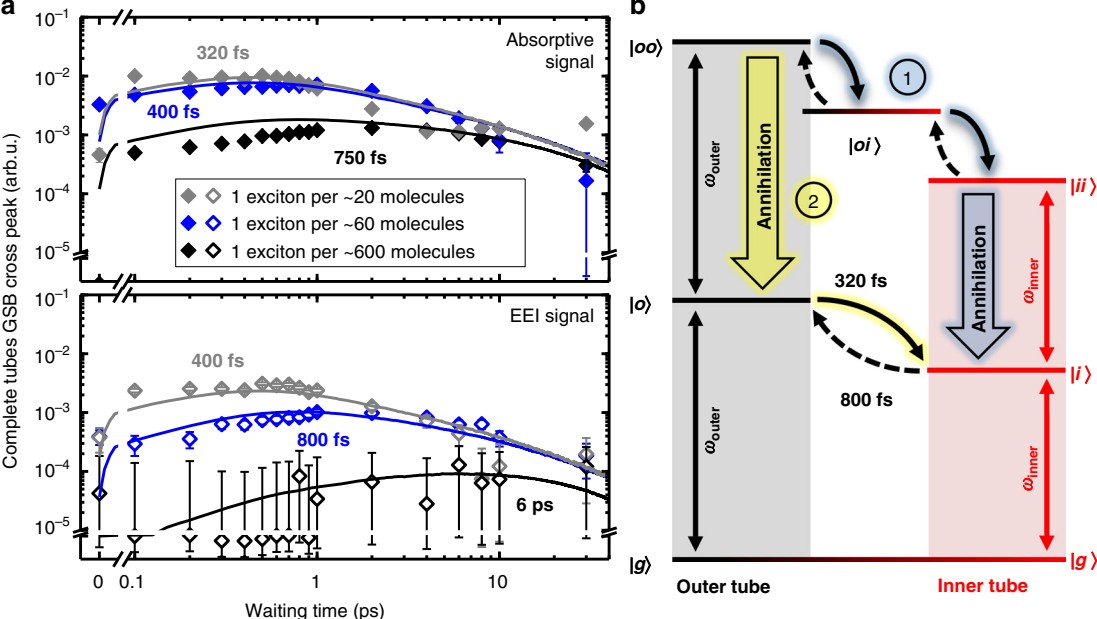

**Fig. 5** Absorptive and EEI cross peak transients with corresponding level diagram. **a** Log–log plot of the absorptive (upper panel, solid diamonds) and EEI (lower panel, open diamonds) GSB/SE transients for the cross peak between outer and inner layer at different exciton densities. The transients were obtained by integrating the signal in the rectangular regions of interest shown in Fig. 2b. The absorptive cross peak maps ET from the outer to the inner tube ($\omega_{outer} \to \omega_{inner}$), while the EEI cross-peak maps the subsequent occurrence of EEA and ET of two excitons from the outer tube ($2\omega_{outer} \to \omega_{inner}$). The amplitude (vertical) scaling is identical to those in Figs. 3 and 4. The error bars refer to the detection noise level in the experiment (Supplementary Note 3). The solid lines depict the results from MC simulations of the exciton dynamics with parameters summarized in Table 1. For each fitting curve the delay time at which the maximum signal occurs is explicitly stated. **b** Energy level diagram of the double-walled nanotubes illustrating bi-exciton (annihilation) pathways 1 (blue) and 2 (green) in presence of both tubes. Optical transitions of the inner and outer tube are marked by vertical arrows and their corresponding frequencies. Curved (dashed) solid arrows depict (thermally activated) ET pathways with their time constants indicated

**Table 1 Overview of parameters for Monte Carlo simulations of the exciton dynamics for isolated inner tubes and complete nanotubes**

| Quantity | Symbol | Inner tubes | Complete nanotubes | Source |
|---|---|---|---|---|
| One-exciton lifetime | $\tau$ | 58 ps | 33 ps | PL measurements; Supplementary Note 9 |
| Annihilation radius | $R_0^{inner}$ | Three molecules | Three molecules | Global fitting parameter; Supplementary Note 6.3 |
| | $R_0^{outer}$ | – | Three molecules | |
| Initial exciton density (number of molecules per exciton) | $\frac{N_m}{N_e}$ | 26 | | Obtained from excitation flux; varied within uncertainty; Supplementary Note 1 |
| | | 57 | 14 | |
| | | 170 | 87 | |
| | | 580 | 853 | |
| Molecular grid size | Inner | $30 \times 1000$ | $30 \times 1000$ | Derived from model in ref. [7]; Supplementary Note 6.1 |
| | Outer | – | $55 \times 1000$ | |
| Lattice constant | $a$ | 0.74 nm | 0.74 nm | Derived from model in ref. [7]; Supplementary Note 6.1 |
| Exciton transfer rate | | | | Obtained from 2D experiments; Supplementary Note 13 |
| (inner → outer) | $k_{io}$ | – | $0.0013\ fs^{-1}$ | |
| (outer → inner) | $k_{oi}$ | – | $0.0031\ fs^{-1}$ | |
| Hopping rate | $H_{inner}$ | $0.04\ fs^{-1}$ | $0.04\ fs^{-1}$ | Global fitting parameter |
| | $H_{outer}$ | – | $0.04\ fs^{-1}$ | |
| Diffusion constant | $D_{2D}$ | $10\ mol\ ps^{-1}$ | $10\ mol\ ps^{-1}$ | Exciton mean square displacement; Supplementary Note 6.4 |
| | | $5.5\ nm^2\ ps^{-1}$ | $5.5\ nm^2\ ps^{-1}$ | |

highlighted in green). Whether (1) or (2) is the prevalent annihilation channel is determined by the balance between the ET and EEA rates. Note that the particular order of ET and EEA during the population time is spectroscopically not distinguishable by examining the cross peak dynamics alone. However, in combination with the respective dynamics of the EEI diagonal peaks a conclusive picture of individual exciton trajectories is obtained.

At the lowest exciton density, a delayed emergence of the EEI cross peak with a maximum at ~6 ps is observed (Fig. 5a, black). In this regime the EEA rate is significantly lower than the ET rate so that the timescale of signal formation is consistent with the EEI signal of the isolated inner tubes. Taken together with the negligibly small EEI signal of the outer tube at this exciton density (Fig. 4a, black) this proves that excitons are harvested by the outer tube and rapidly transferred to the inner tube, where they diffuse and eventually decay, either naturally or via EEA. Therefore, the inner tube acts as an exciton accumulator, which behaves in close analogy to natural systems, where excitation transport is directed via spatio-energetic tuning of the corresponding sites[34,56,57].

At intermediate exciton densities, the vast majority of the EEA events occurs on the outer tube, which is evident from a steep rise of the EEI signal of the outer tube (Fig. 4a), while the inner layer accumulates the already-reduced population of the surviving excitons for which EEA is less pronounced. As a result, the EEI cross peak dynamics are reminiscent to those of the (almost) annihilation-free absorptive cross peak due to balancing of the ET and EEA rates (Fig. 5a, blue; Supplementary Note 6.6 and Supplementary Fig. 14).

For the highest exciton density, the EEA rate exceeds the ET rate. Consequently, the exciton population of the outer tube becomes strongly depleted by EEA prior to any ET. Simultaneously, a significant share of the excitons is transferred to the inner tube resulting in the emergence of the EEI cross peak for which the bottleneck of the rise time is given by the ET rate. In addition, the occurrence of multi-exciton processes gains significance and further reduces the exciton population of the outer tube beyond the two-exciton annihilation picture (Supplementary Notes 6.6 and 7 and Supplementary Figs. 15 and 16), which drastically lowers the fraction of excitons that could be transferred to the inner tube. As a result, the EEI cross peak maximum further shifts toward earlier waiting times (Fig. 5a,

gray), while the amplitude of both absorptive and EEI cross peaks saturates thereby indicating the loss of excitons and, thus, a lower number of transfer events. In the limiting case of instantaneous annihilation of all excitons residing on the outer tube, the formation of the cross peak would be entirely inhibited. In such a way, for increasing excitation fluences the outer tube transitions from an exciton supplying regime into an annihilation regime in which the outer tube exciton population is strongly depleted prior to any transfer to the inner tube.

In order to analyze the observed exciton dynamics, we extend the MC simulations to the case of complete nanotubes. A second layer was added to the molecular grid to represent the outer tube in which the grid size is larger than that of the inner layer in accordance with the increased diameter of the outer tube. The exciton density for the inner and outer tube was set identical (Supplementary Note 1). The excitons are allowed to switch between the adjacent (unoccupied) molecules on the inner and outer layer at the rates specified in the Methods section. Otherwise all parameters are kept identical from the simulations of the isolated inner tubes except the one-exciton lifetime that was measured as 33 ps (Supplementary Note 9 and Supplementary Fig. 19). We extract the absorptive and EEI signals from the MC simulations by evaluating the number of excitons that meet a certain set of prerequisites (Supplementary Table 3). For example, the EEI cross peak ($2\omega_{outer} \rightarrow \omega_{inner}$) is computed as the number of excitons that have been (1) originally planted on the outer tube, (2) participated in at least one annihilation event with an exciton from the same tube, and (3) reside on the inner tube at time $T$. We find excellent agreement between experimental data (symbols) and simulations (solid lines) in Fig. 4 and Fig. 5a by applying the same model parameters for the exciton diffusion and annihilation radius as for the isolated inner tube with exception of the inter-layer ET.

In order to test the exciton diffusion result obtained from our experiments and MC simulation, we also calculated the exciton diffusion constant tensor of C8S3 nanotubes using an extended version of the Haken–Strobl–Reineker model[15,58–60]; see Methods section and Supplementary Note 10.1. From the calculation, we obtained the diffusion constant along the axial direction equal to $23.9\ nm^2\ ps^{-1}$ for the inner wall and $16.3\ nm^2\ ps^{-1}$ for the outer wall of the C8S3 double-walled tube (Supplementary Note 10.2 and Supplementary Table 4). Taken together with a

surface density of 1.8 molecules nm$^{-2}$, where each site contains a unit cell with two molecules, this translates into 43 and 29 molecules ps$^{-1}$ for the inner and outer wall, respectively. These values agree reasonably well with the results obtained from combined experiment and MC simulations of 10 molecules ps$^{-1}$ for both tubes, considering the simplicity of the underlying model for the MC simulations.

Previous measurements of the exciton diffusion constants of supramolecular nanostructures revealed typical values on the order of 100 nm$^2$ ps$^{-1}$ at room temperature assuming purely one-dimensional exciton diffusion[9,16,61], although higher values up to 300–600 nm$^2$ ps$^{-1}$ and even 5500 nm$^2$ ps$^{-1}$ have also been reported[11,62]. These diffusion constants are usually estimated to fall between the limiting cases of fully coherent and purely diffusive transport and, thus, should be considered as an effective diffusion constant with contributions from both processes. Note that it was not possible to obtain a good fit of the experimental data for a purely diffusive model with the diffusion constant increased to 100 nm$^2$ ps$^{-1}$ (Supplementary Note 11 and Supplementary Fig. 21).

**Exciton transfer regimes**. Figure 6 summarizes the main findings of this work as a plot of exciton transfer efficiency versus exciton density. At low exciton densities, the transfer efficiency converges to the value of ~0.7, which is determined by the condition that the exciton populations residing on the inner and outer tube eventually reach thermal equilibrium[63,64]; see Methods section. At high exciton densities, the dynamics are dominated by EEA on the outer tube, which substantially reduces the fraction of transferred excitons and, thus, leads to a reduced transfer efficiency. The maximum indicates optimal balancing between a low degree of EEA on the outer layer, fast inter-layer exciton transfer and subsequent annihilation of the transferred excitons on the inner layer (Supplementary Note 6.6 and Supplementary Fig. 14).

**EEA and exciton delocalization**. Finally, we comment briefly on the effect of exciton delocalization on the EEA process. Like exciton transport, EEA can either proceed in a hopping Förster-

like mechanism[51,55,65] or in a wavelike fashion[66]. While the exciton transport is determined by the energies and couplings of the ground-state transitions of individual molecules that also lead to exciton delocalization, EEA involves coupling through higher excited states[67]. Consequently, the phenomena of exciton delocalization and EEA are closely related, but their relationship is not straightforward. The here presented combination of higher order nonlinear spectroscopy and controlled structural complexity has the potential to unravel the connection between exciton transport (be it wavelike or diffusive) and EEA. Clearly, more theoretical support is needed to fully disentangle these processes, as the annihilation may also depend in a non-trivial way on the phases of the wavefunctions of the involved excitons[68].

In conclusion, we have unambiguously identified the excitonic properties of a complex supramolecular system by utilizing a spectroscopic microfluidic approach. Microfluidic flash-dilution allowed manipulating the structural hierarchy of the supramolecular system on the nanoscale via controlled destruction of individual sub-units of the assembly. This provided a direct view on the simplified structure whose spectral response would otherwise have been concealed due to congested spectroscopic features. Assignment of the excitonic properties was performed by employing exciton–exciton-interaction two-dimensional (EEI2D) spectroscopy, which is capable of isolating mutual interactions of individual excitons. Application of this technique to double-walled nanotubes together with extensive theoretical modeling allowed retrieving a unified set of excitonic properties for the exciton diffusion and exciton–exciton interactions for both layers.

In the arrangement of the double-wall nanotubes, the outer layer appears to act as an exciton antenna, which under strong excitation fluences leads to fast EEA rates prior to any inter-layer ET. At low exciton densities, the inner tube acts as an exciton accumulator absorbing the majority of the excitons from the outer layer. In this capacity, our findings shed light on the importance of the multi-layered, hierarchical structure for the functionality of the light-harvesting apparatus in which the already beneficial excitonic properties of individual sub-units are retained in a more complex double-walled assembly. Hence, the excitonic properties of the supramolecular assembly can be considered robust against variations in the inter-layer transport despite the weak electronic coupling between the layers and the lack of inter-layer exciton coherences. Such excitonic robustness paired with fast inter-layer exciton transfer would prove key for efficient exciton transfer in natural chlorosomes due to close similarity of their telescopic structure with the double-wall nanotubes considered herein. Moreover, we envision that the versatility of the microfluidic approach paired with higher order 2D spectroscopy opens the door to further expedite a better fundamental understanding of the excitonic properties of supramolecular assemblies and, thereby, will encompass rational design principles for future applications of such materials in optoelectronic devices.

## Methods

**Materials and sample preparation**. C8S3 nanotubes were prepared via the alcoholic route[8]. The aggregation of the dye molecule 3,3′-bis(2-sulfopropyl)-5,5′,6,6-tetrachloro-1,1′-dioctylbenzimidacarbocyanine (C8S3, $M = 903$ g mol$^{-1}$) purchased from FEW Chemicals GmbH (Wolfen, Germany) into double-walled nanotubes was verified by linear absorption spectroscopy prior to any other experiments. In order to minimize the thermodynamically induced formation of thicker bundles of nanotubes, sample solutions were freshly prepared for every experiment and used within 3 days.

**Steady-state absorption**. Steady-state absorption spectra were recorded using either a PerkinElmer Lambda 900 UV/VIS/NIR or a Jasco V-670 UV–Vis spectrometer. The sample solution was put either in a 200 µm cuvette (Hellma

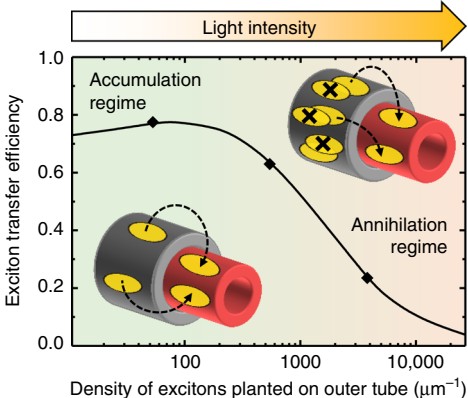

**Fig. 6** Exciton transfer regimes. Exciton transfer efficiency, i.e., fraction of excitons that were planted on the outer tube and either decayed naturally or annihilated on the inner tube as a function of linear exciton density (i.e., the number of excitons per unit of nanotube length), obtained from MC simulations (black line). Symbols indicate exciton densities used in the experiments. In the simulations also the inner tube is populated with excitons at time zero with the same exciton density as the outer tube. The insets schematically depict the exciton (orange ellipses) dynamics in the accumulation regime (bottom left) and the annihilation regime (top right). Dashed arrows: exciton transfer; black crosses: exciton–exciton annihilation

Analytics, Germany) or a 1 mm quartz cuvette (Starna GmbH, Germany). For the latter case, the sample solutions were diluted with Milli-Q water by a dilution factor between 2 and 3.5.

**Microfluidic flash-dilution**. Microfluidic flash-dilution of C8S3 nanotubes was achieved in a tear-drop mixer (micronit, the Netherlands) by mixing neat sample solution with a diluting agent (50:50 mixture of water and methanol by volume) at a flowrate ratio of 5:7. Measurements on the complete nanotubes were conducted by replacing the diluting agent (water and methanol) with Milli-Q water, which only dilutes the sample and does not induce flash-dilution of the outer layer. All solutions were supplied by syringe pumps (New Era, model NE-300). For EEI2D experiments the mixed sample solution was relayed to a transparent thin-bottom microfluidic flow-cell (micronit, the Netherlands) with a channel thickness of 50 μm and a width of 1 mm. With these parameters a maximum optical density of 0.1–0.2 was reached.

**Exciton–exciton interaction 2D (EEI2D) spectroscopy**. More details on the experimental setup are published elsewhere[43]; a schematic of the setup is shown in Supplementary Fig. 22. In brief, the output of a Ti:Sapphire-Laser (Spitfire Pro, Spectra Physics, 1 kHz repetition rate) was focused into a fused-silica hollow-core fiber (UltraFast Innovations) filled with argon to generate a broadband white-light continuum. The main fraction of the light was used as the pump beam and guided through a grism compressor and for further compression through an acousto-optical programmable dispersive filter (DAZZLER, Fastlite, France) to achieve a pulse width of ~15 fs at the sample position (verified via SHG-FROG measurements). The DAZZLER was also used for spectral selection of the excitation spectrum. The remaining fraction of the white-light continuum was used as the probe beam and delayed relative to the pump beam by passing a motorized delay stage (M-IMS600LM, Newport). Both beams were then focused and spatially overlapped in a microfluidic channel under a small angle of 2°. The intensity FWHM of the pump and probe focal spots at the sample position were ~140 μm and ~80 μm, respectively, to minimize the intensity variation of the pump beam over the profile of the probe beam. The polarization of both beams was set parallel to the flow direction of the sample. After passing the sample the spectrum of the probe beam was measured by a CCD camera.

In order to measure 2D spectra the DAZZLER was used to split the pump pulses into two phase-locked time-delayed replica, the delay between which was scanned from 0 fs to 197.6 fs in steps of 0.38 fs. This choice set the resolution along the excitation axis and the Nyquist limit to 84 cm⁻¹ and 44000 cm⁻¹, respectively. The resolution of the probe axis (20 cm⁻¹) was fixed by the detector (ActonSpectraPro 2558i and Pixis 2K camera, Princeton Instruments). In order to isolate the desired 2D signal from unwanted contributions due to background and scattering, the pump and the probe beams were both synchronously modulated by two choppers (MC2000, Thorlabs). All four possible combinations were measured: both beams open, only probe open, only pump open, and both beams blocked. Each contribution was averaged over five consecutive laser pulses by modulating the pump and probe beam at 200 Hz and 100 Hz, respectively. In order to ensure that the spectral region of interest is free of any artifacts from the experimental apparatus, control experiments were performed on an annihilation-free sample (sulforhodamine 101 dissolved in water; Supplementary Note 12 and Supplementary Fig. 23). All experiments were carried out under ambient conditions.

The different data sets of the double-walled nanotubes were measured at pulse energies of the pump pulse of 20, 5, and 0.5 nJ corresponding to exciton densitites of 19 ± 7, 64 ± 23, and 625 ± 228 monomeric units per exciton (Supplementary Note 1). The uncertainty of the exciton density was computed via propagation of uncertainty of all relevant input parameters. For the flash-diluted samples pulse energies of 20, 5, 2.5, and 1 nJ were used corresponding to 18 ± 8, 83 ± 38, 165 ± 75, and 404 ± 185 monomeric units per exciton. The pulse energies were measured at zero time delay of the double pulse.

**Monte Carlo (MC) simulations**. MC simulations of the exciton populations were performed for isolated inner tubes and complete nanotubes represented by a single and two coupled planes, respectively (Fig. 7). Each plane comprised a square grid of molecules with periodic boundary conditions in either direction. The length of the planes was set to 1000 molecules, while the lateral grid size was chosen as 55 molecules (outer tube) and 30 molecules (inner tube) and a lattice constant of 0.74 nm as derived from previously published theoretical models (ref. [7] and Supplementary Note 6.1). For isolated inner tubes, only the inner plane was used. Excitons are depicted as orange circles in order to visualize their annihilation radius. In the MC simulations excitons can perform the following processes: (1) decay according to their lifetime, (2) hop between adjacent sites, (3) vertically transfer between the two layers and (4) undergo EEA. The latter occurred, when two excitons were mutually overlapping within their annihilation radius, as exemplarily shown on the outer layer.

At time zero, excitons were randomly planted on the molecular grid according to the experimental exciton density. Thereafter, the excitons performed a 2D random walk on the grid (with a hopping probability $H$ to move to any of the neighboring molecules) with a time step of 1 fs. In addition, at each step they could be transferred between adjacent molecules on the inner/outer layer or undergo EEA causing the instant deletion of one of the excitons. The latter occurred with a

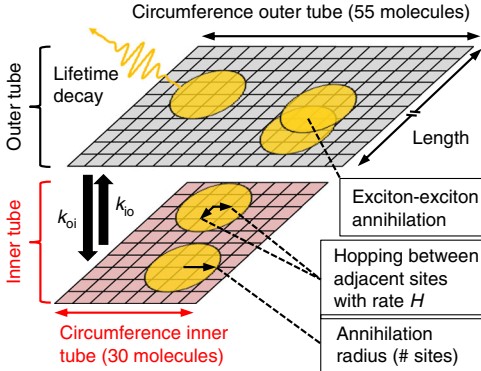

**Fig. 7** Molecular grid for Monte Carlo simulations. The inner and outer tube are depicted as planes shaded in red and gray, respectively. Excitons are shown as orange circles with their size corresponding to the annihilation radius. The different processes that excitons can undergo during the MC simulations are exemplarily shown. For simulations of the exciton dynamics of the isolated inner tubes, only the bottom plane was used

probability of one under the condition that two excitons approach each other closer than the annihilation radius (Supplementary Note 6.3). Excitons were not constrained from (sequential) participation in multiple annihilation events, for which experimental evidence is provided by the observation of higher order signals (Supplementary Note 7). No anisotropic exciton transport (Supplementary Note 10) was included in the MC simulations, but instead the hopping rates were set identical for inner and outer tube in all directions.

In the MC simulations only the exciton hopping rate (i.e., the probability of an exciton to move to any of the neighboring molecules during one time step in the simulation) and the annihilation radius were treated as free parameters, while all other parameters were fixed as their values were obtained from supplementary experiments or calculations. The exciton density was taken from the experimental conditions and allowed to vary within the experimental uncertainty. The lifetime of a single exciton was measured in time-resolved photoluminescence (PL) experiments under extremely low exciton densities of less than 1 exciton per $10^4$ molecules (Supplementary Note 9). The transfer rate from the outer to the inner tube was measured using conventional 2D spectroscopy (Supplementary Note 13 and Supplementary Fig. 24) and agrees with the values from literature[18,69,70]. The opposite rate (inner → outer) follows from the condition that the inner and outer tube exciton populations eventually reach thermal equilibrium, where the net inter-tube transfer rates are identical[63,64]. Hence, this rate is scaled with the Boltzmann factor ($\exp\left(\frac{-\Delta E}{k_B T}\right) \approx 0.22$; with $\Delta E = 300$ cm⁻¹ as the energy difference between inner and outer tube and $k_B T \approx 200$ cm⁻¹ at room temperature) and the density-of-states. The latter is proportional to the number of molecules in the inner and outer layer, which scales with the tube radii assuming identical molecular surface densities (Supplementary Note 6.1). Taken together one finds a ratio of ~0.4 between the upward and the downward exciton transfer rates.

In order to extract the absorptive and EEI signals from the MC simulations, all excitons were labeled with their zero-time position as well as their participation in an annihilation event with an exciton that was originally planted on the same tube. At each time step of the MC simulation the number of excitons was evaluated that met a certain set of prerequisites (Supplementary Table 3). Taking only exciton populations into account (i.e., diagonal entries in a density-matrix description) neglects any possible exciton coherences in the system, which we justify with previously reported findings that any coherence in this system does not survive longer than a few hundred fs[70] and the absence of coherent beatings in the cross peak signal from conventional 2D spectroscopy (Supplementary Fig. 24). For comparison with the experimental results, the simulation transients for the absorptive signals were scaled with identical coefficients to obtain the best fit with experimental data; the same was done for the EEI signals.

**Haken–Strobl–Reineker model**. In order to calculate the exciton diffusion tensor of C8S3 nanotubes, we adopted the same molecular structure for the nanotubes as reported by Eisele et al.[7] The individual tensor elements were then calculated using the following equation:

$$D_{\mathbf{u,w}} = \frac{1}{Z}\sum_{\mu,\nu=1}^{N}\frac{\Gamma}{\Gamma^2 + (\omega_{\mu\nu})}\hat{j}^*_{\mu\nu}(\mathbf{u})\hat{j}_{\mu\nu}(\mathbf{w})\exp\left(\frac{-\hbar\omega_\nu}{k_B T}\right). \tag{1}$$

Here, $\mu$ and $\nu$ run over all the $N$ collective exciton states, obtained by diagonalizing the exciton Hamiltonian for the tube considered (ref. [7] and Supplementary Note 10.1), $\Gamma$ is the dephasing rate that characterizes the Haken–Strobl–Reineker model of white noise thermal fluctuations[15,58–60] and $\hbar\omega_{\mu\nu} = \hbar(\omega_\mu - \omega_\nu)$ is the energy difference between exciton states $\mu$ and $\nu$.

Furthermore, $\hat{j}_{\mu\nu}(\mathbf{u}) = i \sum_{n,m=1}^{N} \langle \mu|m\rangle (\mathbf{u} \cdot \mathbf{r}_{mn}) J_{nm} \langle n|\nu\rangle$ is the flux operator along direction $\mathbf{u}$ in the exciton eigenstate basis, where $n$, $m$ run over all the molecules in the aggregate, $\mathbf{r}_{mn} = \mathbf{r}_m - \mathbf{r}_n$ is the relative separation vector between molecules $m$ and $n$, and $J_{nm}$ is the excitation transfer (dipole-dipole) interaction between them. To describe this interaction, we use extended transition dipoles instead of point dipoles, as this better describes the excitation transfer interactions between nearby molecules. The Boltzmann factor $\exp\left(\frac{-\hbar\omega_\nu}{k_B T}\right)$ is used to account in a simple way for a temperature $T$ smaller than the exciton bandwidth and $Z = \sum_{\nu=1}^{N} \exp\left(\frac{-\hbar\omega_\nu}{k_B T}\right)$ is the exciton partition function. An asterisk (*) on $\hat{j}_{\mu\nu}(\mathbf{u})$ refers to complex conjugation of the operator.

A detailed derivation of the above equation excluding the Boltzmann factor can be found elsewhere[15]. For the C8S3 nanotubes, each wall has a diffusion tensor, characterized by the tensor elements $D_{z,z}$, $D_{z,\phi}$, $D_{\phi,z}$, and $D_{\phi,\phi}$, where $z$ is the axial direction and $\phi$ is the direction along the circumference of the tube. Further details are given in Supplementary Note 10.

## Data availability
The data that support the findings of this study are available from the corresponding author upon request.

## Code availability
The computer code for the Monte Carlo simulations is available through the journal website.

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

## Acknowledgements

B.K. and M.S.P. thanks A. S. Bondarenko, I. Patmanidis, A. H. de Vries, and S. J. Marrink for numerous discussions. F. de Haan is greatly acknowledged for writing the code for Monte Carlo simulations as well as general laboratory assistance. B.K., J.K., T.L.C.J., and M.S.P. acknowledge funding by the Dieptestrategie Programme of the Zernike Institute for Advanced Materials (University of Groningen, the Netherlands). M.S.P. has also received funding from the European Union's Horizon2020 research and innovation programme under Marie Sklodowska Curie Grant No. 722651. T.B. acknowledges funding by the European Research Council (ERC)—Grant No. 614623, the Deutsche Forschungsgemeinschaft (DFG, German Research Foundation)—Grant No. BR 423942615, and the Bavarian State Ministry of Science, Research, and the Arts—Collaborative Research Network "Solar Technologies Go Hybrid".

## Author contributions

B.K. prepared the samples. B.K. and J.L. performed the absorptive and EEI2D experiments and analyzed the experimental data, together with P.M.; the analysis was supervised by T.B. and M.S.P. B.K. performed the Monte Carlo simulations. T.K. calculated the exciton diffusion tensor under supervision of T.L.C.J. and J.K. J.K. led the discussion on the link between exciton delocalization and annihilation. B.K. and M.S.P. wrote the manuscript with contributions from all other authors.

## Competing interests

The authors declare no competing interests.

## Additional information

**Peer Review Information** *Nature Communications* thanks Dongho Kim, Martin Zanni and the other, anonymous, reviewer(s) for their contribution to the peer review of this work. Peer reviewer reports are available.

