## [Peer Review File · Nature Communications]

Reviewers' comments:

Reviewer #1 (Remarks to the Author):

The authors presents an experimental study, supported by simulations, understanding exciton dynamics in an artificial light harvesting complex, comprised of one carbon nanotube embedded in a larger nanotube. Specifically they study exciton diffusion leading to annihilation with exciton-exciton interaction two dimensional spectroscopy. The experiments reveal that the outer layer acts as an exciton antenna while the inner as a sink. The author hypothesise that results from these experiments can offer insight into how exciton dynamics may occur in natural systems - and I am inclined to agree with them.

This is very good work, with extensive results. The question is why have the authors chosen to publish this work as a communication? I found it very difficult to understand the significance of the work without needing to constantly consult the SI, and I am still a little confused about key mechanisms. It made reviewing and reading the paper quite difficult.

My view is that a communication should be self contained, and the SI should there for technical details only. In this case, the authors seem to be using its a way of cutting down the word count of their paper, presumably so that they can publish this work a "high impact" communication. I urge the authors to turn this into a high quality full paper. If feel it is essential that at least, Supplementary Figure 2, some the discussion regarding the double-sided Feynman diagrams and some of the simulation details need to be in the main paper. These aper to be key toothier discussion. There may even be more that should be put in the main paper.

Specific notes that the authors should also try to address;

(1) The referencing seems to be quite restricted to European groups. As a polite request, it would be nice to see a bit more diversity.

(2) Can you explain exactly what is meant by hierarchy? Presumably the fact one nanotube structure is embedded in the other? While the word is used a lot, I'm not sure the definition of what is meant is given.

(3) In Figure 1, you say that as the laser spectrum doesn't cover the band at 520 nm, it won't affect the dynamics. As this looks like it has a long tail, are the authors sure there won't be absorption in this band via multiple photons?

(4) I'm a little confused with their statement at the top of page 9 regarding "uphill energy transfer". Surely there is no such thing? These features must come from excited state absorption?

(5) On page 11 you mention "Pauli repulsion" - can you explain how this works for exciton-exciton interactions, and why Coulombic repulsion is not important.

(6) On page 12, you speak of your system having both one- and two-dimensional character. Is it possible to get a quasi-quantitative picture by extrapolating between the two?

(7) On page 15, and in the SI, you speak of dipole-dipole interactions. These assume small (vanishing) molecular size compared to the optical wavelengths. Is that still the case for these systems, or should you include a multipolar formalism?

Overall, this is very nice work, and I am sorry if I disappoint the authors, insisting that this would be better a full paper - but unless the authors can somehow restructure this to include the key points I address above, moving more details from the SI, into the main body, I feel it shouldn't be rewritten up as a communication. I do however think it would make an excellent full paper.

Reviewer #2 (Remarks to the Author):

This manuscript written by Maxim and Brixner groups touches on an important topic of exciton-exciton annihilation (EEA) and energy/exciton transfer (ET) dynamics in a double-walled tubular J-aggregate which consists of the amphiphilic cyanine dye, C8S3. Actually, the exciton dynamics of the tubular C8S3 aggregate studied by two-dimensional electronic spectroscopy is already been reported in the literature (J. Phys. Chem. A, 2010, 114, 8179. the authors cite this paper in the manuscript.). However, this work is distinguished by the novelty of the following points: 1) they use a state-of-the-art exciton-exciton interaction two-dimensional electronic spectroscopy (EEI2D, Nat. Commun., 2018, 9, 2466.) which is a pertinent technique to disentangle the multi-exciton (more than two) interactions from the one-exciton dynamics. 2) By adapting microfluidics in the time-resolved spectroscopy, they can simultaneously investigate the excited-state dynamics both in a

single-walled inner tube and a double-walled complete tube. And with the aid of theoretical modelings, the authors finally conclude that the inner and outer tubes act as an exciton accumulator and exciton antenna, respectively. In my opinion, this work is well organized and all the experimental and calculation data draw attention to the presented conclusion. As such this paper will be of interest to readership of Nat. Commun. and therefore publication is recommended after a minor revision.

Minor

1. Although C8S3 is well-known molecule, it would be better to include molecular structures in Figure 1 to help readers understand.

Reviewer #3 (Remarks to the Author):

The authors have presented a manuscript employing a unique combination of micro-fluidic devices with multi-dimensional spectroscopy techniques to measure the transient dynamics of excitons within supramolecular nanotubes. I feel that the demonstration of the combination of these experimental techniques is unique. Implementation of microfluidics in this case appears to allow a high degree of control over the supramolecular structure of the system (inner+outer shells vs only inner shell) from which comparisons of the exciton dynamics can be made. The authors do an excellent job utilizing multidimensional spectroscopy to reveal the energy transfer pathways in the nanotubes studied, and further provide complementary modeling to develop a more complete portrait of the system dynamics.

Overall I found the experimental description and the analysis of the data quite thorough. One criticism that I would point out is some of the language regarding how the outer shell protects the inner tube from 'over-burning' at high fluences. First, 'over-burning' is an imprecise term that is used without any real explanation of what process it represents. The wavelengths of the absorption of the outer and inner shells are different, so I don't see any evidence that the presence of the out-shell in any way 'protects' the inner shell from absorption of excitons. It seems that the presence of the outer-shell can only add additional excitons to the inner shell via exciton transfer. Even if exciton from the outer to inner shell is limited at higher fluences, the inner shell should still be absorbing a high density of excitons at its absorption frequency. I don't believe the authors present any evidence of 'burning' or other damage mechanisms take place with the absence of the outer-shell, so I would be inclined to request that the authors clarify their discussion regarding these claims in the manuscript.

To this end, I also feel that the manuscript could benefit from a more direct comparison between the dynamics of the inner shell alone with the combined outer+inner shell. The two dynamics plots within the paper show the diagonal peak for the inner peak and the cross-peak for the outer+inner peak. The Author motivates the analysis of the cross-peak by stating that the absorption by the outer tube is large, however, based on the fact that they are absorbing at differing wavelengths I do not believe that the outer tube is shielding the inner tube from absorption. If this is the case, I would then be interested to see a comparison of the diagonal peak dynamics of the inner tube with and without the presence of the outer tube. Would EEA processes on the diagonal be slower if inner tube excitons are preferentially annihilating with excitons from the outer tube?

Overall, I would recommend this manuscript for publication after minor revisions to address the comments above have been made.

First of all, we would like to thank the Editor and Reviewers for their time and efforts that have helped us to improve the manuscript. We address all raised issues below and in a revised version of the manuscript.

Reviewers' comments:

Reviewer #1 (Remarks to the Author):

The authors presents an experimental study, supported by simulations, understanding exciton dynamics in an artificial light harvesting complex, comprised of one carbon nanotube embedded in a larger nanotube. Specifically they study exciton diffusion leading to annihilation with exciton-exciton interaction two dimensional spectroscopy. The experiments reveal that the outer layer acts as an exciton antenna while the inner as a sink. The author hypothesise that results from these experiments can offer insight into how exciton dynamics may occur in natural systems - and I am inclined to agree with them.

This is very good work, with extensive results. The question is why have the authors chosen to publish this work as a communication? I found it very difficult to understand the significance of the work without needing to constantly consult the SI, and I am still a little confused about key mechanisms. It made reviewing and reading the paper quite difficult.

My view is that a communication should be self contained, and the SI should there for technical details only. In this case, the authors seem to be using its a way of cutting down the word count of their paper, presumably so that they can publish this work a "high impact" communication. I urge the authors to turn this into a high quality full paper. If feel it is essential that at least, Supplementary Figure 2, some the discussion regarding the double-sided Feynman diagrams and some of the simulation details need to be in the main paper. These aper to be key toothier discussion. There may even be more that should be put in the main paper.

Answer: In response to the comment, we have introduced the requested changes in the manuscript:

- We have merged Figure 2 from the main text with Supplementary Figure 2 and Supplementary Figure 3 in order to show the complete evolution of the spectra for different waiting times, all now contained in the main manuscript. We chose to show the spectra recorded under high exciton densities in order to highlight all features present in the spectra, i.e., the presence of diagonal and cross peaks in both the absorptive and the EEI regions of the spectra. The spectra for low exciton densities are still shown in the SI for completeness.
- For the discussion of the double-sided Feynman diagrams, we have added a selection of diagrams to Figure 3 in a new panel c, together with a detailed description in the Figure caption and in the main text.
- We have moved a substantial amount of Supplementary Information on the MC simulation to the Method section of the main paper. This includes the section on the exchange rates

between the inner and outer tube, the overview of all parameters as well as the schematic of the molecular grid for MC simulations (formerly Supplementary Figure 10; now Figure 7).

Despite these additions, we stayed well below the maximum word count set forth by the journal policy.

The remainder of Supplementary Information is either technical or expert-specific or describing control experiments so that it is not directly relevant to the main story (but of course is very relevant for verifying accuracy and correctness of our measurements and interpretations). Thus we decided not to move these contents.

Specific notes that the authors should also try to address;

(1) The referencing seems to be quite restricted to European groups. As a polite request, it would be nice to see a bit more diversity.

Answer: We have adopted a more balanced diversity in our references by replacing a number of previously used references, but keeping in mind the limit of 70 references:

- References #4, #6, #25, #26, #27, #33 and #40 in the original manuscript were removed in order to accommodate new references from other groups (see below) and additional references due to shifting part of the SI to the main text.
- References #12, #23, #28, #29, #36, #41 and #52 in the revised manuscript have been introduced to diversify the origin of the involved research groups.
- The conference proceeding in reference #47 in the original manuscript was replaced by a recently published peer-reviewed paper (now #44).

We note, however, that initially we selected the references on the basis of their relevance to the current work but not geographical location.

(2) Can you explain exactly what is meant by hierarchy? Presumably the fact one nanotube structure is embedded in the other? While the word is used a lot, I'm not sure the definition of what is meant is given.

Answer: We agree with the reviewer that we have not formally introduced this term, which may have led to confusion. The term hierarchy here refers to the structural hierarchy (as mentioned in the title) of the supramolecular nanotubes under study, i.e., a clear distinction of inner and outer tube as sub-units of the complete assembly. Another level of hierarchical exists via the emergence of one tube from self-assembly of the constituent molecules. We have reformulated two sentences in the introduction in the following way:

- (cf. p. 3; introduction) "Understanding the origin of the delocalized states and tracking energy transport throughout the entire complex hierarchical structure of multi-chromophoric systems – from the individual molecules, over individual sub-units all the way up to the complete multi-layered assembly – is vital to unravel nature's highly successful design principles."

- (cf. p. 4; introduction): “Recent studies have focused on reducing the complexity of multi-layered, supramolecular nanotubes and thereby essentially uncoupling individual hierarchical units, i.e., the inner and outer layer of the assembly by oxidation chemistry.”

(3) In Figure 1, you say that as the laser spectrum doesn't cover the band at 520 nm, it won't affect the dynamics. As this looks like it has a long tail, are the authors sure there won't be absorption in this band via multiple photons?

Answer: We have carefully inspected the spectral regions in the 2D spectra (for flash-diluted nanotubes under the highest excitation fluence) where the signal of C8S3 monomers (peaking at ~540 nm) could have appeared under one- and two-photon excitation. In neither case we found a signal that would resemble the expected response from C8S3 monomers from which we conclude that the excitation and response of monomers are irrelevant for our analysis. This agrees very well with estimation of exciton densities of the C8S3 monomers and isolated inner tubes (1 exciton per ~3300 and ~20 molecules, respectively).

To clarify this issue for the reader, we have added a more elaborate discussion of this issue to the SI (cf. p. 7ff): “Supplementary Note 2: C8S3 Monomers Signal via One-and Two-Photon Absorption”.

(4) I'm a little confused with their statement at the top of page 9 regarding “uphill energy transfer”. Surely there is no such thing? These features must come from excited state absorption?

Answer: What is meant here is that there may be thermally-activated exciton transfer from the inner (low energy; ~600 nm) to the outer tube (high energy; ~590 nm). In other words, at room temperature excitons residing on the inner tube have a finite probability to transfer to the energetically higher-lying states belonging to the outer tube as dictated by the canonical detailed balance condition, which leads to the Boltzmann probability distribution in thermal equilibrium. We have described this in further detail in the context of the exciton transfer rates in thermal equilibrium in the method section (cf. p. 32).

In order to avoid confusion, we have reformulated the following sentences

- (cf. p. 10): “A cross peak above the diagonal can also be identified; however, it has a low amplitude because of thermally activated uphill ($\Delta E \approx 300 \text{ cm}^{-1}$) energy transfer from the inner to the outer tube and its partial spectral overlap with ESA of the inner tube.”
- (cf. p. 20): “Curved solid (dashed) arrows depict downhill (thermally activated uphill) ET pathways with their time constants indicated.”
- We have also stated explicitly that all experiments were conducted at room temperature (cf. p. 29).

(5) On page 11 you mention "Pauli repulsion" - can you explain how this works for exciton-exciton interactions, and why Coulombic repulsion is not important.

Answer: In the main text Pauli repulsion between excitons is mentioned in the context that two excitons cannot reside on the same molecule, which is responsible for the fact that in J-aggregates the lowest two-exciton state has a higher energy than twice the energy of the lowest one-exciton state. In J-aggregates, this generally is the dominant effect in transient absorption, as is well established; see, e.g., Fidler *et al.* (1993) (Ref. 50 in the original manuscript) and Bakalis & Knoester (1999) (Ref. 51 in the original manuscript); Coulomb interactions may contribute to repulsion or attraction between excitons, but will do so only for molecules with permanent dipoles (or considerable higher-order multipoles) in the ground or excited state of the charge-neutral Frenkel exciton.

In order to clarify this issue we have reformulated the corresponding sentence in the manuscript in the following way (cf. p. 13): "For J-aggregates, the latter occurs blue-shifted relative to the ground-state to one-exciton transition ($\Delta > 0$) as a consequence of Pauli repulsion between excitons [REF. 49], as two excitations cannot reside on the same molecule. This effective repulsion between Frenkel excitons dominates Coulomb interactions between them if the difference in the permanent dipole between the ground and excited states considered is zero."

(6) On page 12, you speak of your system having both one- and two-dimensional character. Is it possible to get a quasi-quantitative picture by extrapolating between the two?

Answer: We are not aware of any work related to the extrapolation of the analytical solutions for diffusion-assisted exciton-exciton annihilation beyond the quasi one- and two-dimensional cases. Moreover, the quantum diffusion in such systems is an important, but not the only relevant, aspect for exciton-exciton annihilation. In [Phys. Rev. Lett. **116**, 196803 (2016)] and [JACS **136**, 8963 (2014)] relations for the diffusion constant that interpolate between the 1D and 2D case is presented (albeit for a simple square lattice rolled on a cylinder). In the present study we apply that method in order to verify the diffusion constant extracted from our MC simulations and find good agreement. We do agree that this issue presents an interesting topic, but may require an entirely new study in which the diameter and therefore the effective dimensionality of the system is varied, which is beyond the scope of the current work.

(7) On page 15, and in the SI, you speak of dipole-dipole interactions. These assume small (vanishing) molecular size compared to the optical wavelengths. Is that still the case for these systems, or should you include a multipolar formalism?

Answer: The intermolecular interactions are calculated using extended transition dipoles on each molecule, as explained in the SI (Supplementary Note 11.1). This method is an improvement over the point-dipole interaction, because the distances between neighboring molecules is not much larger than the size of the individual molecules. Using extended dipoles effectively takes into account multipolar effects, as the referee correctly suggests to be of importance. At any rate, whether point-dipoles or extended dipoles are used, the dominant interwall interactions are much weaker than the dominant

intra-wall interactions. (J. Phys. Chem. Lett. 2017, 8, 2895-2901; Supporting Information: subsection 4.5).

In order to clarify the use of extended transition dipoles to the reader we have added the following sentences to the Method section (cf. p. 34): “To describe this interaction, we use extended transition dipoles instead of point dipoles, as this better describes the excitation transfer interactions between nearby molecules.”

Overall, this is very nice work, and I am sorry if I disappoint the authors, insisting that this would be better a full paper - but unless the authors can somehow restructure this to include the key points I address above, moving more details from the SI, into the main body, I feel it shouldn't rewritten up as a communication. I do however think it would make an excellent full paper.

Answer: No disappointment from the authors – on the contrary, we think that moving the material from the SI to the main text has strengthened the manuscript!

Reviewer #2 (Remarks to the Author):

This manuscript written by Maxim and Brixner groups touches on an important topic of exciton-exciton annihilation (EEA) and energy/exciton transfer (ET) dynamics in a double-walled tubular J-aggregate which consists of the amphiphilic cyanine dye, C8S3. Actually, the exciton dynamics of the tubular C8S3 aggregate studied by two-dimensional electronic spectroscopy is already been reported in the literature (J. Phys. Chem. A, 2010, 114, 8179. the authors cite this paper in the manuscript.). However, this work is distinguished by the novelty of the following points: 1) they use a state-of-the-art exciton-exciton interaction two-dimensional electronic spectroscopy (EEI2D, Nat. Commun., 2018, 9, 2466.) which is a pertinent technique to disentangle the multi-exciton (more than two) interactions from the one-exciton dynamics. 2) By adapting microfluidics in the time-resolved spectroscopy, they can simultaneously investigate the excited-state dynamics both in a single-walled inner tube and a double-walled complete tube. And with the aid of theoretical modelings, the authors finally conclude that the inner and outer tubes act as an exciton accumulator and exciton antenna, respectively. In my opinion, this work is well organized and all the experimental and calculation data draw attention to the presented conclusion. As such this paper will be of interest to readership of Nat. Commun. and therefore publication is recommended after a minor revision.

Answer: We are pleased to read that the reviewer considers our findings interesting, convincing and of interest to the broad community of Nature Communications.

Minor

1. Although C8S3 is well-known molecule, it would be better to include molecular structures in Figure 1 to help readers understand.

Answer: We have added the molecular structure to Figure 1 in an additional, new panel a (cf. p. 7, Figure 1).

Reviewer #3 (Remarks to the Author):

The authors have presented a manuscript employing a unique combination of micro-fluidic devices with multi-dimensional spectroscopy techniques to measure the transient dynamics of excitons within supramolecular nanotubes. I feel that the demonstration of the combination of these experimental techniques is unique. Implementation of microfluidics in this case appears to allow a high degree of control over the supramolecular structure of the system (inner+outer shells vs only inner shell) from which comparisons of the exciton dynamics can be made. The authors do an excellent job utilizing multidimensional spectroscopy to reveal the energy transfer pathways in the nanotubes studied, and further provide complementary modeling to develop a more complete portrait of the system dynamics.

Overall I found the experimental description and the analysis of the data quite thorough. One criticism that I would point out is some of the language regarding how the outer shell protects the inner tube from 'over-burning' at high fluences. First, 'over-burning' is an imprecise term that is used without any real explanation of what process it represents. The wavelengths of the absorption of the outer and inner shells are different, so I don't see any evidence that the presence of the out-shell in any way 'protects' the inner shell from absorption of excitons. It seems that the presence of the outer-shell can only add additional excitons to the inner shell via exciton transfer. Even if exciton from the outer to inner shell is limited at higher fluences, the inner shell should still be absorbing a high density of excitons at its absorption frequency. I don't believe the authors present any evidence of 'burning' or other damage mechanisms take place with the absence of the outer-shell, so I would be inclined to request that the authors clarify their discussion regarding these claims in the manuscript.

Answer: We agree with the reviewer that the language on "over-burning" was not clear in the original manuscript. The reviewer has correctly pointed out that the number of excitons that the inner tube receives does not decrease in absolute terms, but only relative to the number of excitons placed on the outer tube. Our intention was to convey the message how the outer layer changes its functionality from supplying excitons to the inner tube under low fluences to an "annihilation" regime at high fluences for which the EEA rate outcompetes the ET rate.

In order to avoid this confusion we have carefully reformulated the following sentences in the abstract and in the manuscript:

- (cf. p. 2; abstract): "We show that at low excitation fluences, the outer layer acts as an exciton antenna that supplies excitons to the inner tube, while under high excitation fluences the outer layer converts its functionality into an exciton annihilator where the exciton population is depleted prior to any exciton transfer to the inner tube."
- (cf. p. 21f; main text): "In such a way, for increasing excitation fluences the outer tube transitions from an exciton supplying regime into an annihilation regime in which the outer tube exciton population is strongly depleted prior to any transfer to the inner tube."

To this end, I also feel that the manuscript could benefit from a more direct comparison between the dynamics of the inner shell alone with the combined outer+inner shell. The two dynamics plots within the paper show the diagonal peak for the inner peak and the cross-peak for the outer+inner peak. The Author motivates the analysis of the cross-peak by stating that the absorption by the outer tube is large, however, based on the fact that they are absorbing at differing wavelengths I do not believe that the outer tube is shielding the inner tube from absorption. If this is the case, I would then be interested to see a comparison of the diagonal peak dynamics of the inner tube with and without the presence of the outer tube. Would EEA processes on the diagonal be slower if inner tube excitons are preferentially annihilating with excitons from the outer tube?

Answer: As the reviewer has correctly pointed out the absorption peaks of the inner and outer tubes are at different wavelengths and, thus, absorption by the outer tube does not directly affect absorption by the inner tube and *vice versa*. In the original manuscript this point might have been confusing, as the outer tube does not actually take a protective role in the sense outlined by the reviewer, but we used this language to emphasize the change in functionality of the outer tube transitioning from an exciton supplying to an exciton annihilating regime. In the revised manuscript, we have clarified the language used. Also, in order to facilitate the direct comparison between the complete system and the isolated inner tubes for the reader we have shifted the figure containing the inner and outer tube diagonal peak transients for the complete system to the main text (formerly Supplementary Figure 17, now Figure 4).

Overall, I would recommend this manuscript for publication after minor revisions to address the comments above have been made.

REVIEWERS' COMMENTS:

Reviewer #1 (Remarks to the Author):

The authors have responded to my changes satisfactorily and the work is now publishable. It looks like a nice paper!

However regarding my point 4; I still think "uphill energy transfer" is an unfortunate term, even qualifying it as "thermally activated". In dipole-dipole coupled resonance energy transfer (RET), the actual transfer event can only be resonant - a virtual photon is transferred between donor and acceptor transitions that are in both in resonance with the mediating virtual photon (hence the R in RET). The fact that the process may start from hot vibronic states doesn't change the fact that the actual RET process is resonant. Theoretically, if you bring phonon modes from the bath into the system Hamiltonian then we would only see a resonant process. The term of "uphill energy transfer" to me is very misleading. Thermally activated chemical reactivity is on a PES is something different (which I guess is what the authors are seeing in conjunction with RET).

I won't insist the authors take the term "uphill" out (it will be up to them to justify this in future), but I personally wouldn't use the term.

Reviewer #3 (Remarks to the Author):

Thank you for taking the time to make these small changes in the wording of the manuscript following my brief points in the initial review. Based on these editorial changes, and those corresponding to the other reviewers I would recommend this manuscript for publication with no further edits.

REVIEWERS' COMMENTS:

Reviewer #1 (Remarks to the Author):

The authors have responded to my changes satisfactorily and the work is now publishable. It looks like a nice paper!

However regarding my point 4; I still think "uphill energy transfer" is an unfortunate term, even qualifying it as "thermally activated". In dipole-dipole coupled resonance energy transfer (RET), the actual transfer event can only be resonant - a virtual photon is transferred between donor and acceptor transitions that are in both in resonance with the mediating virtual photon (hence the R in RET). The fact that the process may start from hot vibronic states doesn't change the fact that the actual RET process is resonant. Theoretically, if you bring phonon modes from the bath into the system Hamiltonian then we would only see a resonant process. The term of "uphill energy transfer" to me is very misleading. Thermally activated chemical reactivity is on a PES is something different (which I guess is what the authors are seeing in conjunction with RET).

I won't insist the authors take the term "uphill" out (it will be up to them to justify this in future), but I personally wouldn't use the term.

Answer: We understand the referee's concern and omit the term 'uphill' in our manuscript in order to avoid any confusion. Instead we refer to it as 'thermally activated' energy transfer as the referee suggested (cf. p. 10 and p. 20).

Reviewer #3 (Remarks to the Author):

Thank you for taking the time to make these small changes in the wording of the manuscript following my brief points in the initial review. Based on these editorial changes, and those corresponding to the other reviewers I would recommend this manuscript for publication with no further edits.

Answer: We thank the referee their effort and time.